

# Physiological and molecular response mechanisms of tomato seedlings to cadmium (Cd) and lead (Pb) stress

Yan Zhou[1,2,3], Jinyu Fu[1], Yuqi Ye[1], Qibo Xu[1], Jinjie Liang[1], Yanyan Chen[1], Yuxing Mo[1,2,3] and Kaidong Liu[1,2,3]

[1] Lingnan Normal University, Zhanjiang, China
[2] GuangDong Technology Innovation Center of Tropical Characteristic Plant Resource Development, Lingnan Normal University, Zhanjiang, Guangdong, China
[3] Zhanjiang Key Laboratory of Tropical Characteristic Plant Technology Development, Lingnan Normal University, Zhanjiang, Guangdong, China

Corresponding authors
Yan Zhou, 286138826@qq.com
Kaidong Liu,
liukaidong2001@126.com

## ABSTRACT

Heavy metal contamination, particularly from cadmium (Cd) and lead (Pb), poses significant risks to soil and water resources and leads to severe damage in plants. This study investigated the physiological and molecular mechanisms of the responses of tomato (*Solanum lycopersicum* L.) seedlings to Cd and Pb stress by applying 50 mg/L Cd, 100 mg/L Pb, and a combination of 50 mg/L Cd + 100 mg/L Pb. The goal was to understand how these heavy metals impact the growth, antioxidant systems, and secondary metabolic pathways in tomato seedlings. The results showed that compared with the control, Cd + Pb stress significantly increased the content of soluble sugar by 37.40% and 33.46% on days 5 and 15, respectively, and the content of proline by 77.91% to 93.91% during the entire period in tomato seedlings. It also elevated electrical leakage by 110.52% on day 15, maintained the levels of malondialdehyde close to the control, enhanced the activities of superoxide dismutase by 33.32% on day 10 and 11.22% on day 15, peroxidase by 42.15% on day 10, and catalase by 90.78% on day 10. Additionally, it reduced the contents of hydrogen peroxide by 15.47% to 29.64% and the rate of formation of superoxide anions by 26.34% to 53.47% during the entire period of treatment. The transcriptomic analysis revealed a significant differential expression of the genes involved in pathways, such as phenylalanine, glutathione, arginine and proline, and nitrogen metabolism. These genes included *PALs*, *HDCs*, *GGCT*, *ODC1*, *LAPs*, *SMS*, and *SAMDC*. Notably, transcription factors, such as *ERF109*, *ARF9*, *GRF3*, *GRF4*, *GRF7*, and *GRF9*, were also significantly regulated. The study concluded that Cd and Pb stress enhanced the osmoregulatory and antioxidant defense systems in tomato seedlings, which may contribute to their tolerance to heavy metal stress. Future research could explore the application of these findings to develop strategies to improve the resistance of plants to contamination with heavy metals.

## INTRODUCTION

Substantial amounts of agricultural, industrial, and sewage waste, which contain heavy metals and other pollutants that significantly infiltrate the soil, are indiscriminately discharged into the environment (*Gola et al., 2016*; *Mohanty & Das, 2023*). These discharges lead to severe environmental pollution issues and exacerbate the levels of contamination of soil and water bodies, which further contributes to the overall deterioration of the environment (*Ibrahim, Ibrahim & Yusuf, 2021*). Agricultural practices, industrial activities, and improper waste disposal are the key sources of cadmium (Cd) and lead (Pb) contamination in soil (*Mohanty & Das, 2023*). The Agency for Toxic Substances and Disease Registry (ATSDR) lists Pb and Cd among the top 275 toxic substances to humans (*Yuan et al., 2015*). Exposure to Pb or Cd poses significant harm to human health, including long-term negative effects on neurological development in children's IQ and learning disabilities and a decline in memory and cognitive functions in adults (*Bellinger, 2008*). Additionally, exposure to Cd causes kidney dysfunction and renal failure (*Johri, Jacquillet & Unwin, 2010*). Heavy metal toxicity, particularly the non-essential heavy metals, exerts significant adverse effects on plant growth and development even at low concentrations (*Shah et al., 2010*) by interfering with the plants' normal physiological and biochemical processes, which affects the expression of genes (*Atkinson & Urwin, 2012*). Moreover, heavy metals reduce the biosynthesis of carbohydrates and nucleic acids and disrupt essential processes, such as cell division and metabolism, that are crucial for the growth and reproduction of plants (*Peralta-Videa et al., 2009*). This cascade of effects leads to stunted plant growth and abnormal development, which lowers the overall quality of agricultural products (*Hatfield & Prueger, 2015*). This phenomenon impacts agricultural output and indirectly threatens human health and food safety by introducing harmful metals into the food chain.

Notably, various plant species have evolved a considerable degree of tolerance to heavy metals (*Shah et al., 2010*). Plants have evolved and acquired a suite of molecular and physiological mechanisms to evade or detoxify heavy metals, including the activation of antioxidant defense systems to scavenge reactive oxygen species (ROS), an increase in the production of specific secondary metabolites, and the modulation of the signaling transduction pathways (*Manara, 2012*; *Tang et al., 2023*). The ingress of substantial amounts of Cd and Pb into plant tissues causes severe physiological damage, primarily because of the overproduction and accumulation of ROS, which disrupts the selective permeability of the plasma membrane and affects intracellular substance exosmosis and electrical leakage (*Ali, Tyagi & Bae, 2023*; *Ur Rahman et al., 2024*). Plants counteract the overproduction and accumulation of ROS by activating NADPH oxidases to produce hydrogen peroxide ($H_2O_2$) and scavenge excess ROS through antioxidant enzymes, such as superoxide dismutase (SOD) and ascorbate peroxidase (APX). Plants also utilize antioxidants, such as proline, polyphenol oxidase (PPO), and glutathione (GSH), to alleviate oxidative damage and maintain a cellular redox balance (*Hasanuzzaman et al., 2012*; *Rajput et al., 2021*). The AsA-GSH cycle is a vital pathway for the removal of ROS, which protects the plants from Cd or Pb damage by increasing the activities of antioxidant

enzyme and the content of antioxidants (*Zhang, 2021*). These mechanisms demonstrate the complex adaptive mechanisms of plants to single metal stress (*Tavanti et al., 2021*). Plants also increase the production of specific secondary metabolites and modulate signaling transduction pathways to adapt to heavy metal stress from Cd and Pb (*Khare et al., 2020*). Activation of the phenylalanine metabolic pathway is one such critical response because phenylalanine is a precursor for various secondary metabolites related to defense (*Yadav et al., 2020*). Enhanced metabolic activity promotes the biosynthesis of antioxidants and detoxifying compounds, such as anthocyanins and lignin, which are crucial for resisting the oxidative stress induced by heavy metals (*Sytar et al., 2013*). Moreover, the metabolism of glutathione directly participates in scavenging free radicals and facilitates the excretion of heavy metals by forming chelates, thereby reducing intracellular toxicity. Arginine and proline metabolism also significantly influence the plant response mechanisms (*Anjum et al., 2011*). Arginine is a direct precursor of proline and participates in its biosynthesis, thereby promoting its accumulation within the cells (*Szabados & Savouré, 2010*). Proline acts as an osmoprotectant, maintains cellular osmotic balance and mitigates the osmotic pressure stress caused by heavy metals, thus, protecting the plant cells from damage (*Sharma, Schat & Vooijs, 1998*). Arginine also serves as a precursor to nitric oxide (NO) and enhances its biosynthesis. This regulates plant stress responses and activates antioxidant defense systems, thereby mitigating the toxicity of heavy metals (*Khan et al., 2023*). Arginine directly participates in adjusting the metabolic responses of plants to heavy metal stress and indirectly enhances plant tolerance and defense capabilities by regulating signaling molecules through these complex and crucial pathways.

Plants enhance their resistance to heavy metal stress by finely regulating the pathways of nitrogen metabolism. Nitrogen regulation is primarily manifested in the optimization of the biosynthesis of specific amino acids, such as increasing the levels of the precursor amino acids glutamate, glycine, and cysteine, for the antioxidant GSH and augmenting the levels of proline and arginine (*Ali et al., 2020*). These amino acids maintain the cellular osmotic balance, alleviate osmotic pressure stress, and enhance the detoxification capacity and antioxidant defense system by participating in or promoting antioxidant reactions and signaling transduction (*Sharma & Dietz, 2006*). Moreover, the adjustment of nitrogen metabolism supports the biosynthesis of stress-responsive proteins, such as metal-binding proteins and antioxidant enzymes, which further enhance resistance to heavy metal stress (*Hall, 2002*). Plants effectively manage heavy metal stress and protect themselves from toxicity through these complex regulatory mechanisms (*Sytar et al., 2013*). Transcription factors (TFs) play a pivotal role in regulating the responses of plants to stress. They recognize and bind to the promoter regions of specific genes, thereby modulating their expression (*Yoon et al., 2020*). Several families of TFs, including *WRKY*, *bZIP*, *NAC*, *AP2/ERF*, *MYB*, and *GRF*, are activated under Cd and Pb stress (*Meraj et al., 2020*). These TFs induce antioxidative defenses, the chelation of metal ions, and excretion by activating or inhibiting the expression of the metal transporter proteins, antioxidant enzymes, metal chelators, and genes associated with signaling transduction, thereby enhancing the tolerance of plants to heavy metal stress (*Hasan et al., 2017*). The TFs strongly link

metabolic pathways with the regulation of gene expression and form an effective network system to combat heavy metal stress (*Hall, 2002*). In summary, the response of plants to Cd and Pb stress involves a complex and efficient physiological and molecular regulatory network formed during the long course of evolution. Plants mitigate the toxic effects of heavy metals and maintain stability in growth, development, and metabolism by regulating crucial pathways and factors, such as phenylalanine, glutathione, arginine, proline, and nitrogen metabolism and TFs (*Viehweger, 2014*; *Hoque et al., 2021*; *Zheng et al., 2023*). Thus, a deep understanding of these mechanisms is a prerequisite for future endeavors to develop and utilize plants for the remediation of heavy metal pollution.

Tomato (*Solanum lycopersicum* L.) is a crop of economic significance that is used extensively in research on plant defenses against biotic and abiotic stresses (*Kimura & Sinha, 2008*). The use of tomatoes to remediate the soil degradation caused by contamination with heavy metals to ensure safe crop production is a novel phytoremediation strategy (*Aggarwal & Goyal, 2007*). Currently, most studies have primarily focused on the tolerance of tomatoes to single stresses of Cd or Pb, while the tolerance and genetic mechanisms that underlie the response of tomatoes to combined Cd and Pb stresses remain unclear. Herein, we conducted a study to perform the following: (1) examine the phenotypic, physiological, and transcriptional changes in tomatoes upon exposure to Pb and Cd; (2) identify the genes that respond to Cd and Pb, as well as the pathways associated with phenylalanine, glutathione, arginine, proline, and nitrogen metabolism; and (3) comprehensively characterize the physiological tolerance and genetic mechanisms of tomatoes in response to combined Pb and Cd stresses. These findings offer new insights into the complex molecular mechanisms and stress response pathways activated by tomatoes, which will contribute to a deeper understanding of the resilience of plants to multiple heavy metal stressors. These results lay the groundwork for future research and potential applications in phytoremediation strategies to mitigate soil contamination.

## MATERIALS AND METHODS

### Plant materials and experimental treatments

'Xinzhongshu No. 4' tomatoes were used as the experimental plants in this study. The seeds were germinated and then sown within a substrate blend that was composed of vermiculite, perlite, and peat moss in equal proportions (1:1:1, v/v/v). Uniformly growing tomato seedlings at the two-leaf stage were subsequently transplanted into cultivation vessels that contained the mixed substrate described in the previous statement and maintained in a controlled growth environment with a photoperiod of 12 h light (30,000 lux/25 °C), 10 h night (0 lux/17 °C), and 70% to 80% humidity. Distinct treatments were administered through root irrigation when the seedlings had matured to the four-leaf stage by irrigating each vessel with 50 mL of treatment solution.

The experiment was composed of the following four treatments: (1) Control: distilled water; (2) Cd: 50 mg/L cadmium chloride ($CdCl_2$); (3) Pb: 100 mg/L lead acetate ($PbAc_2$); and (4) Cd + Pb: a combination of 50 mg/L $CdCl_2$ and 100 mg/L $PbAc_2$. A randomized complete block design was employed, which featured three replicates per treatment and 21

plants per treatment. This resulted in a total of 84 plants. The concentrations of Cd and Pb used were determined through preliminary experiments (Tables S1 and S2; Figs. S1 and S2). Samples were obtained at intervals of 5, 10, and 15 days post-treatment to assess the pertinent physiological parameters. The transcriptomic analysis was conducted 15 days post-treatment.

## Growth parameters

The growth indicators of tomato seedlings, including the plant height (distance from the stem base to the growing point), stem diameter, leaf length, and leaf width, were measured at 0, 5, 10, and 15 days post-treatment. Increment values were calculated by determining the difference between each parameter at each treatment period and its respective value at the preceding treatment period (Zhou et al., 2022a). The aboveground and belowground parts of the seedlings were collected separately, and the surface debris was removed with distilled water. The seedlings were then rinsed thoroughly with deionized water and dried with a cloth. The fresh weight of each sample was recorded. The samples were then inactivated at 105 °C for 15 min followed by drying at 75 °C until a constant weight was achieved. The dry weight was then recorded.

## Physiological indicators

Electrical leakage was measured using the conductance method. The contents of proline and soluble sugars were quantified using the acid ninhydrin method (Ábrahám et al., 2010) and the anthrone colorimetric method (Alagbe et al., 2020), respectively. The content of malondialdehyde (MDA) was determined using the thiobarbituric acid reactive substances assay (Zhou et al., 2017). The content of $H_2O_2$ was determined as previously described (Zhou et al., 2023a). The contents of superoxide anions ($O_2{}^{\cdot-}$) were assessed through hydroxylamine oxidation (Kozuleva, 2022). SOD, peroxidase (POD), catalase (CAT), and APX were assayed using kits according to the manufacturer's instructions (Suzhou Kemeing Biotechnology Co., Ltd., Suzhou, China).

## RNA isolation and sequencing

The total RNA of 0.1 g of tissue per sample was extracted using an EASYspin Plant RNA Kit (Aidlab, Beijing, China) followed by an assessment of RNA integrity, concentration, and quality using a NanoDrop spectrophotometer (LifeReal, Hangzhou, China) and an Agilent 2100 instrument (Agilent Technologies, Santa Clara, CA, USA). High-quality RNA (1.5 µg) characterized by the absence of smears in the agarose gel electrophoresis (AGE) and with RNA Integrity Number (RIN) values ≥8.0, $A_{260/280}$ ratios that ranged between 1.8 and 2.1, and $A_{260/230}$ ratios ≥2.0 were obtained (Zhou et al., 2023b). There were three biological replicates. The statistical power of this experimental design was 0.84 owing to the calculations in RNASeqPower. The cDNA library was constructed and sequenced as previously described (Zhou et al., 2022b). The raw sequence data obtained were deposited in the National Center for Biotechnology Information (NCBI) database under accession number PRJNA1085857.

### Quality control, read mapping, and enrichment analysis of the differentially expressed genes (DEGs)

The raw paired-end reads were trimmed using the FASTP tool (https://github.com/OpenGene/fastp) set at default to obtain the clean reads. The resulting clean reads were independently aligned to the reference genomes using HISAT2 (http://ccb.jhu.edu/software/hisat2/index.shtml) configured in the orientation mode. Mapped reads from each sample were assembled using StringTie (https://ccb.jhu.edu/software/stringtie/). The levels of expression of the individual genes were assessed using the transcripts per million (TPM) read approach, which further revealed the DEGs between the comparison groups. RSEM (https://deweylab.biostat.wisc.edu/rsem/) was utilized to quantify the abundance of genes. DEGs with a |log2(fold change)| ≥ 1 and $P < 0.05$ were considered statistically significant. The DEGs were functionally enriched using the Gene Ontology (GO) (http://www.geneontology.org) and Kyoto Encyclopedia of Genes and Genomes (KEGG) (http://www.genome.jp/kegg/) databases to assess whether the DEGs were significantly enriched in the GO terms and metabolic pathways. The significance threshold was set at a $P$-adjusted ≤ 0.05 level compared to the background of the whole transcriptome. Goatools (https://github.com/tanghaibao/Goatools) and KOBAS (https://bio.tools/kobas) were employed for the GO functional enrichment and KEGG pathway analyses.

### Quantitative reverse transcription PCR analysis

Eight candidate genes were precisely selected from the transcriptomic dataset for validation using quantitative reverse transcription polymerase chain reaction (qRT-PCR) (*Zhou et al., 2022b*). Primer sequences (Table S3) for the qRT-PCR amplification were designed using Primer Premier version 5.0. The relative levels of expression of the DEGs identified across the distinct samples were quantified using the delta-delta Ct ($2^{-\Delta\Delta Ct}$) method as described by *Livak & Schmittgen (2001)*. The housekeeping gene actin was used as the internal standard to determine the relative expression of the DEGs.

### Statistical analysis

Statistical analyses were conducted using a one-way analysis of variance (ANOVA) in SPSS 19.0 (IBM, Inc., Armonk, NY, USA) by separately analyzing the data from each experimental group. The statistical significance was determined using the Tukey's *post hoc* test at a significance threshold of $P < 0.05$. Distinct alphabetical letters were used to signify significant differences between the experimental groups. The experimental design was completely randomized with three biological replicate for all treatments.

## RESULTS

### Growth parameters

Compared with the control, the Cd + Pb treatment significantly increased the height of tomato seedlings by 33.64% and 130.00% on days 10 and 15 post-treatment, respectively (Table 1). The Cd, Pb, and Cd + Pb treatments significantly enhanced the stem diameter by 355.56%, 230.00%, and 204.17%, respectively, on day 15 post-treatment. The Pb treatment significantly decreased the width of the leaves of the tomato seedlings by 90.00% at day 15.

**Table 1 Growth indexed in Cd- and Pb-stressed tomato seedlings.**

| Treatment | Time | The increment of plant height (cm) | The increment of stem diameter (mm) | The increment of leaf length (cm) | The increment of leaf width (cm) | Leaf fresh weight (g) | Leaf dry weight (g) | Root fresh weight (g) | Root dry weight (g) |
|---|---|---|---|---|---|---|---|---|---|
| Control | 5 d | 0.85 ± 0.05c | 0.260 ± 0.07c | 0.90 ± 0.10a | 0.33 ± 0.05a | 17.67 ± 0.61a | 1.90 ± 0.07a | 4.53 ± 0.16a | 0.48 ± 0.02a |
| Cd | | 1.75 ± 0.15a | 0.493 ± 0.08a | 0.70 ± 0.00b | 0.18 ± 0.08c | 15.20 ± 0.10b | 1.63 ± 0.01b | 3.90 ± 0.03b | 0.41 ± 0.01b |
| Pb | | 1.25 ± 0.05b | 0.310 ± 0.06bc | 0.27 ± 0.12c | 0.23 ± 0.03bc | 12.20 ± 0.10c | 1.33 ± 0.02c | 3.14 ± 0.01c | 0.33 ± 0.01c |
| Cd + Pb | | 1.10 ± 0.10bc | 0.392 ± 0.06ab | 0.67 ± 0.06b | 0.30 ± 0.00ab | 10.37 ± 0.15d | 1.15 ± 0.04d | 2.67 ± 0.02d | 0.29 ± 0.01d |
| Control | 10 d | 1.10 ± 0.10b | 0.030 ± 0.00c | 0.28 ± 0.10bc | 0.13 ± 0.06ab | 25.67 ± 0.61a | 2.52 ± 0.06a | 6.58 ± 0.16a | 0.70 ± 0.02a |
| Cd | | 1.43 ± 0.21a | 0.237 ± 0.03a | 0.42 ± 0.03a | 0.15 ± 0.06a | 23.20 ± 0.10b | 2.28 ± 0.01b | 5.95 ± 0.02b | 0.64 ± 0.01b |
| Pb | | 1.03 ± 0.06b | 0.165 ± 0.02b | 0.17 ± 0.06c | 0.05 ± 0.05b | 20.20 ± 0.10c | 1.98 ± 0.01c | 5.18 ± 0.03c | 0.55 ± 0.01c |
| Cd + Pb | | 1.47 ± 0.06a | 0.210 ± 0.01ab | 0.33 ± 0.05ab | 0.10 ± 0.00ab | 18.37 ± 0.15d | 1.80 ± 0.01d | 4.74 ± 0.02d | 0.51 ± 0.01d |
| Control | 15 d | 0.50 ± 0.10c | 0.060 ± 0.02c | 0.11 ± 0.03ab | 0.10 ± 0.00ab | 30.67 ± 0.61a | 2.79 ± 0.06a | 7.86 ± 0.16a | 0.84 ± 0.02a |
| Cd | | 0.95 ± 0.05b | 0.273 ± 0.06a | 0.10 ± 0.00b | 0.05 ± 0.05bc | 25.20 ± 0.10b | 2.11 ± 0.01c | 5.96 ± 0.03c | 0.64 ± 0.01c |
| Pb | | 0.63 ± 0.06c | 0.198 ± 0.03b | 0.13 ± 0.07ab | 0.01 ± 0.02c | 23.20 ± 0.10c | 2.29 ± 0.01b | 6.45 ± 0.04b | 0.69 ± 0.01b |
| Cd + Pb | | 1.15 ± 0.05a | 0.183 ± 0.04b | 0.20 ± 0.07a | 0.14 ± 0.05a | 20.37 ± 0.15d | 1.85 ± 0.01d | 5.22 ± 0.04d | 0.56 ± 0.01d |

**Note:**
Values are means ± SD ($n = 3$). Values with a different letter within a sampling date are significantly different ($P < 0.05$).

In contrast, the Cd + Pb treatment significantly increased the width of the leaves on day 15 post-treatment. Compared with the control, the Cd, Pb, and Cd + Pb treatments significantly decreased the leaf fresh weight by 9.61% to 17.83%, 21.30% to 30.94%, and 28.44% to 41.32%, respectively; the leaf dry weight by 9.48% to 24.35%, 17.82% to 30.07%, and 28.30% to 39.56%, respectively; the root fresh weight by 9.56% to 24.26%, 17.95% to 30.72%, and 27.94% to 41.10%, respectively; and the root dry weight by 9.13% to 24.00%, 17.42% to 30.94%, and 27.01% to 40.63%, respectively, throughout the entire treatment period of the tomato seedlings.

## Electrical leakage and the contents of proline and soluble sugar

Compared with the control, the Cd, Pb, and Cd + Pb treatments significantly increased the electrical leakage by 17.84%, 137.95%, and 110.52% on day 15 of post-treatment, respectively. Notably, the Cd + Pb treatment significantly decreased the electrical leakage of the tomato seedlings by 11.53% on day 15 compared with the Pb treatment (Fig. 1). Compared with the control, the Cd treatment significantly decreased the contents of proline and soluble sugar of the tomato seedlings by 66.31% to 89.59% and 9.68% to 49.78%, respectively, during the entire treatment period. Similarly, the Pb treatment significantly decreased the content of proline of the tomato seedlings by 82.68% to 94.60% during the treatment period and significantly decreased the content of soluble sugar by 48.98% and 10.27% on days 5 and 15, respectively. In contrast, the Pb treatment significantly increased the content of soluble sugar by 11.76% on day 10. The Cd + Pb treatment significantly decreased the content of proline in the tomato seedlings by 77.91% to 93.91% during the entire treatment period and the content of soluble sugar by 37.40% and 33.46% on days 5 and 15, respectively.

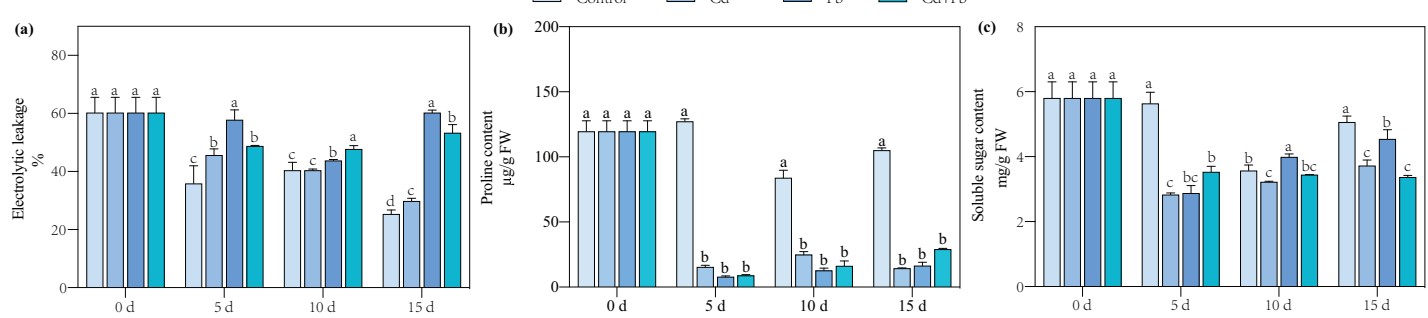

**Figure 1 Electrolytic leakage (A), proline content (B), and soluble sugar content (C) in leaves of Cd- and Pb-stressed tomato seedlings.** Each value is the mean ± standard error ($n = 3$), and the error bars represent the standard error. Bars with a different letter within a sampling date are significantly different ($P < 0.05$).

## Contents of MDA, $H_2O_2$, and $O_2^{\cdot-}$

Compared with the control, the Cd + Pb treatment significantly decreased the contents of $O_2^{\cdot-}$ and $H_2O_2$ of the tomato seedlings by 26.34% to 53.47% and 15.47% to 29.64%, respectively, during the entire treatment period (Fig. 2). In contrast, it increased the content of MDA in the tomato seedlings by 4.56% to 20.17% during the entire treatment period. The Cd + Pb treatment significantly decreased the content of MDA in the tomato seedlings by 15.52% and 15.64% on days 10 and 15 of treatment, respectively, compared with the Cd treatment. Additionally, the Cd + Pb treatment significantly decreased the contents of $O_2^{\cdot-}$ in the tomato seedlings by 12.76% to 26.80% during the entire treatment period and the content of $H_2O_2$ by 10.52% on day 15. In contrast, the Cd + Pb treatment significantly increased the content of MDA in the tomato seedlings by 15.21% and 22.47% on days 5 and 15, respectively, and significantly decreased the content of $O_2^{\cdot-}$ of the tomato seedlings by 28.25% and 25.25% on days 10 and 15, respectively, compared with the Pb treatment.

## Activities of SOD, POD, CAT, and APX

Compared with the control, the Cd treatment significantly increased the SOD activity of the tomato seedlings by 29.07% and 19.13% on days 10 and 15, respectively (Fig. 3). The Pb treatment resulted in a significant overall increase in the activity of SOD in the tomato seedlings by 21.08% to 24.89% during the entire treatment period, while the Cd + Pb treatment significantly increased the activity of SOD in the tomato seedlings by 33.32% and 11.22% on days 10 and 15 compared with the control, respectively. Moreover, the activities of POD and CAT increased significantly by 42.15% and 90.78%, respectively, on day 10 of treatment, while the activity of APX significantly decreased by 34.40% and 40.41% on days 5 and 10 when treated with Cd + Pb compared with the control, respectively. Compared with the Cd treatment, the Cd + Pb treatment significantly increased the activity of POD in the tomato seedlings by 23.86% and 15.24% on days 10 and 15, respectively, but significantly decreased the activity of APX by 46.07% on day 5. Compared with the Pb treatment, the Cd + Pb treatment significantly decreased the

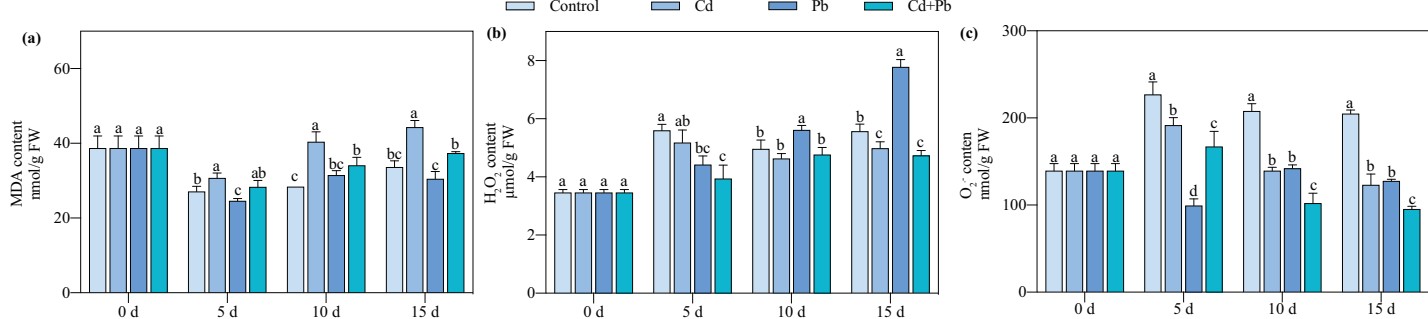

**Figure 2  MDA (A), H₂O₂ (B), and O₂·⁻ (C) in leaves of Cd- and Pb-stressed tomato seedlings.** Each value is the mean ± standard error ($n = 3$), and the error bars represent the standard error. Bars with a different letter within a sampling date are significantly different ($P < 0.05$).

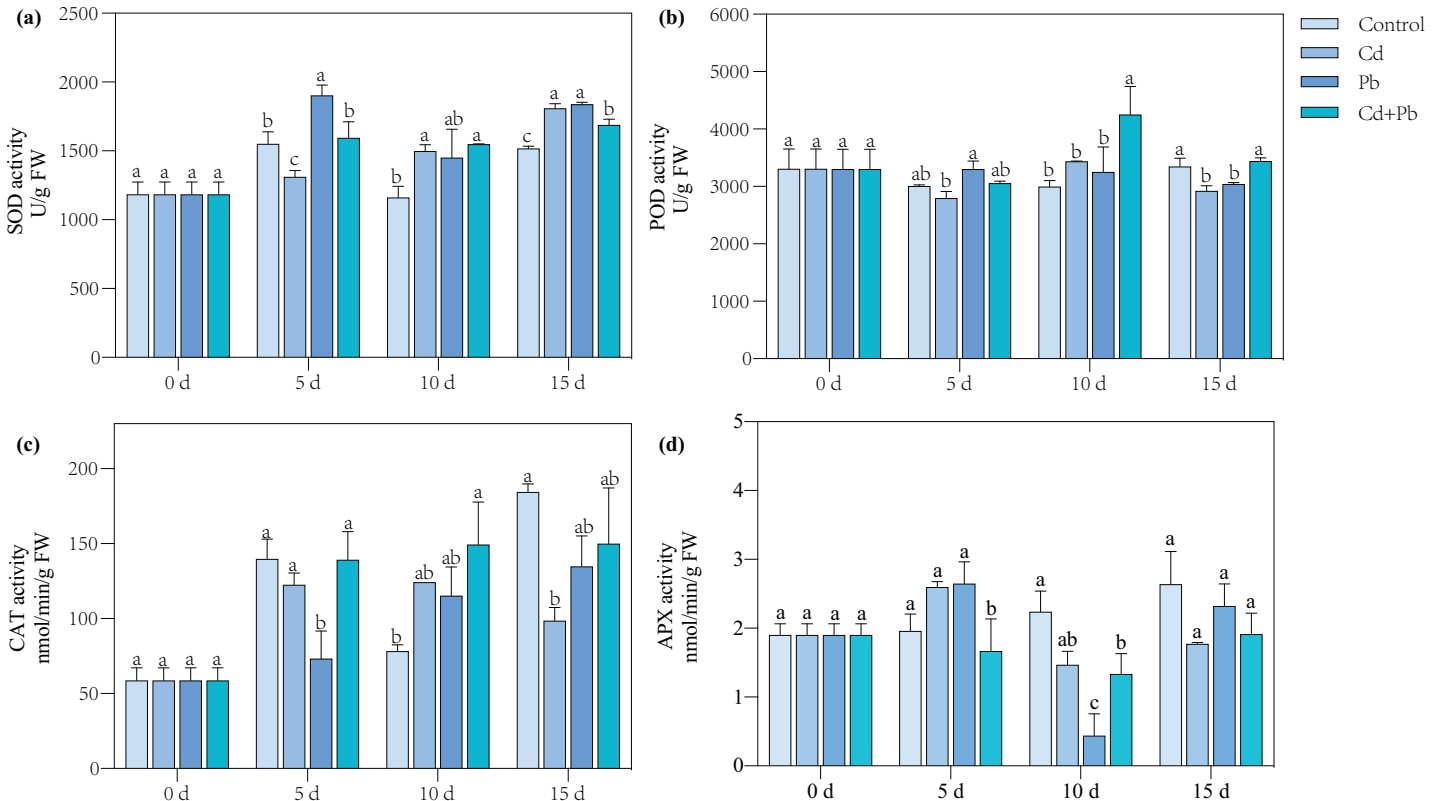

**Figure 3  SOD (A), POD (B), CAT (C), and APX (D) in leaves of Cd- and Pb-stressed tomato seedlings.** Each value is the mean ± standard error ($n = 3$), and the error bars represent the standard error. Bars with a different letter within a sampling date are significantly different ($P < 0.05$).

activity of SOD in the leaves of the tomato seedlings by 16.14% and 8.15% on days 5 and 15, respectively, and it significantly increased the activity of POD by 30.85% and 11.60% on days 10 and 15, respectively. The activity of CAT significantly increased by 90.00% on day 5, while the activity of APX significantly decreased and increased by 36.91% and 66.93% on days 5 and 10, respectively.

## Analysis of the transcriptome profiles and DEGs of the tomato seedlings under Pb and Cd stress

The transcriptome sequencing revealed the profiles of gene expression of the tomato seedlings under the control and Cd, Pb, and Cd + Pb stress after 15 days of treatment. The high-throughput sequencing of 12 libraries yielded 43.67–72.83 million raw reads. Trimming of the low-quality reads yielded clean reads that ranged between 98.59% and 99.20% of the raw reads, with an average of 97.23% mapped reads. Moreover, the Q30 values ranged between 95.35% and 96.20% (Table S4), which suggested that the filtered sequencing data qualified for further analysis.

A differential expression analysis was conducted through a pairwise comparison of the DEGs in the three treatment groups using the control samples as the reference to identify the specific changes in gene expression (Fig. 4). There were 681 upregulated and 556 downregulated DEGs in the Cd *vs.* Control group, 581 upregulated and 531 downregulated DEGs in the Pd *vs.* Control group, 1,340 upregulated and 934 downregulated DEGs in the Cd + Pb *vs.* Control group, 350 upregulated and 331 downregulated DEGs in the Cd + Pb *vs.* Pb group, and 715 upregulated and 484 downregulated DEGs in the Cd + Pb *vs.* Cd group, respectively (Fig. 4A). Moreover, there were 236, 376, and 1,286 unique DEGs in the Pb *vs.* Control, Cd *vs.* Control, and Cd + Pb *vs.* Control groups, respectively (Fig. 4B), while 453 DEGs were commonly regulated in the three pairs of groups. Similarly, there were 236 and 818 unique DEGs in the Cd + Pb *vs.* Pb and Cd + Pb *vs.* Cd groups, respectively, while 381 DEGs were commonly regulated in both pairs of groups.

The expression of the eight DEGs involved in the pathways induced by Cd and Pb was analyzed using qRT-PCR to confirm the validity of transcriptomic profiling (Fig. S3). Notably, the patterns of expression of these DEGs in the leaves of the tomato seedlings in the control, Cd, Pb, and Cd + Pb treatments were consistent with the transcriptome data. This consistency indicated that the RNA-seq data was highly reproducible and reliable.

## Gene co-expression clusters and GO and KEGG pathway enrichment analyses

Compared with the control, 209 genes (Cluster 1) were upregulated, while 150 genes (Cluster 2) were downregulated in the leaves of tomato seedlings under Cd stress (Fig. 5; Dataset S1). The GO enrichment analysis revealed that the upregulated genes were associated with photosystem I (GO:0009522), tetrapyrrole binding (GO:0046906), and monooxygenase activity (GO:0004497), while the downregulated genes were involved in photosynthetic electron transport in photosystem II (GO:0009772), electron transporter (GO:0045156), and protein kinase binding (GO:0019901). Compared to the control, 88 genes (Cluster 1) were upregulated, while 120 genes (Cluster 2) were downregulated in the leaves of tomato seedlings under Pb stress (Fig. 6; Dataset S1). The GO enrichment analysis revealed that the upregulated genes were associated with protein phosphorylation (GO: 0006468), plasma membrane (GO:0005886), and phosphorylation (GO:0016310), while the downregulated genes were involved in xyloglucan: xyloglucosyl transferase activity (GO:0016762), hormone biosynthetic process (GO:0042446), and xyloglucan metabolic process (GO:0010411). In a similar manner, 818 genes (Cluster 1) were upregulated, while

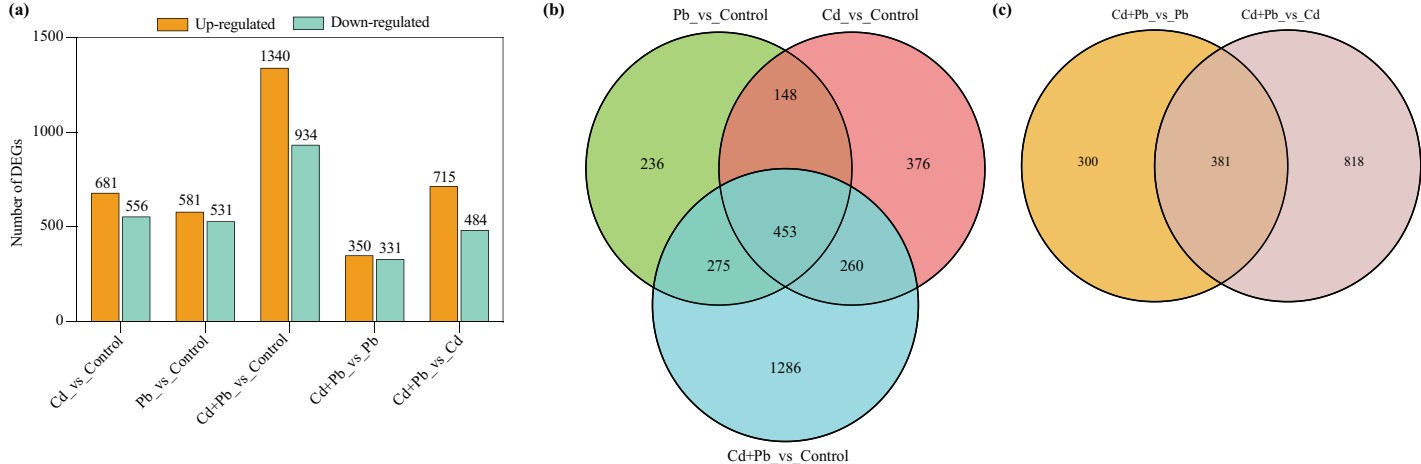

**Figure 4** (A) Up- and down-regulated DEGs in leaves of Cd- and Pb-stressed tomato seedlings on the 15th day. (B) Venn Diagrams depicting DEGs in Cd *vs.* Control group, Pd *vs.* Control group, and Cd + Pd *vs.* Control group. (C) Venn Diagrams depicting DEGs in Cd + Pb *vs.* Pb group, and Cd + Pb *vs.* Cd group.

438 genes (Cluster 2) were downregulated in the leaves of the tomato seedlings under Cd + Pb stress compared to the control (Fig. 7; Dataset S1). The GO enrichment analysis revealed that the upregulated genes were associated with cytoskeletal motor activity (GO: 0003774), microtubule-based movement (GO:0007018), and microtubule motor activity (GO:0003777), while the downregulated genes were involved in the plasma membrane (GO:0005886), integral component of membrane (GO:0016021), and intrinsic component of membrane (GO:0031224).

A GO enrichment analysis was subsequently performed to identify the functions of the common DEGs induced by Cd (453) and Pb (381) stress in tomato seedlings (Figs. 8A and 8C; Dataset S2). The most abundant GO terms of the DEGs induced by Cd stress were response to wounding (GO:0009611) in biological process and serine-type endopeptidase inhibitor activity (GO:0004867) in molecular function (Fig. 8). The most abundant GO terms of the DEGs induced by Pb stress were microtubule-based movement (GO:0007018) in biological process, plasma membrane (GO:0005886) in cellular component, and serine-type endopeptidase inhibitor activity (GO:0004867) in molecular function. The DEGs in response to Cd and Pb were subsequently mapped on the KEGG pathway database. Notably, the most significant pathways of the 453 DEGs induced by Cd stress were glutathione metabolism, phenylalanine metabolism, and diterpenoid biosynthesis (Figs. 8B and 8D; Dataset S3). The common genes between the Cd + Pb *vs.* Pb and Cd + Pb *vs.* Cd group pairs were involved in arginine and proline, nitrogen, and galactose metabolism, which represented the top three significant pathways. All these genes and those involved in glutathione, phenylalanine, arginine and proline, nitrogen metabolism, and the regulation of TFs were analyzed in more detail. The 453 and 381 common DEGs in the Cd + Pb *vs.* Pb and Cd + Pb *vs.* Cd group pairs, respectively, are listed in Dataset S3.

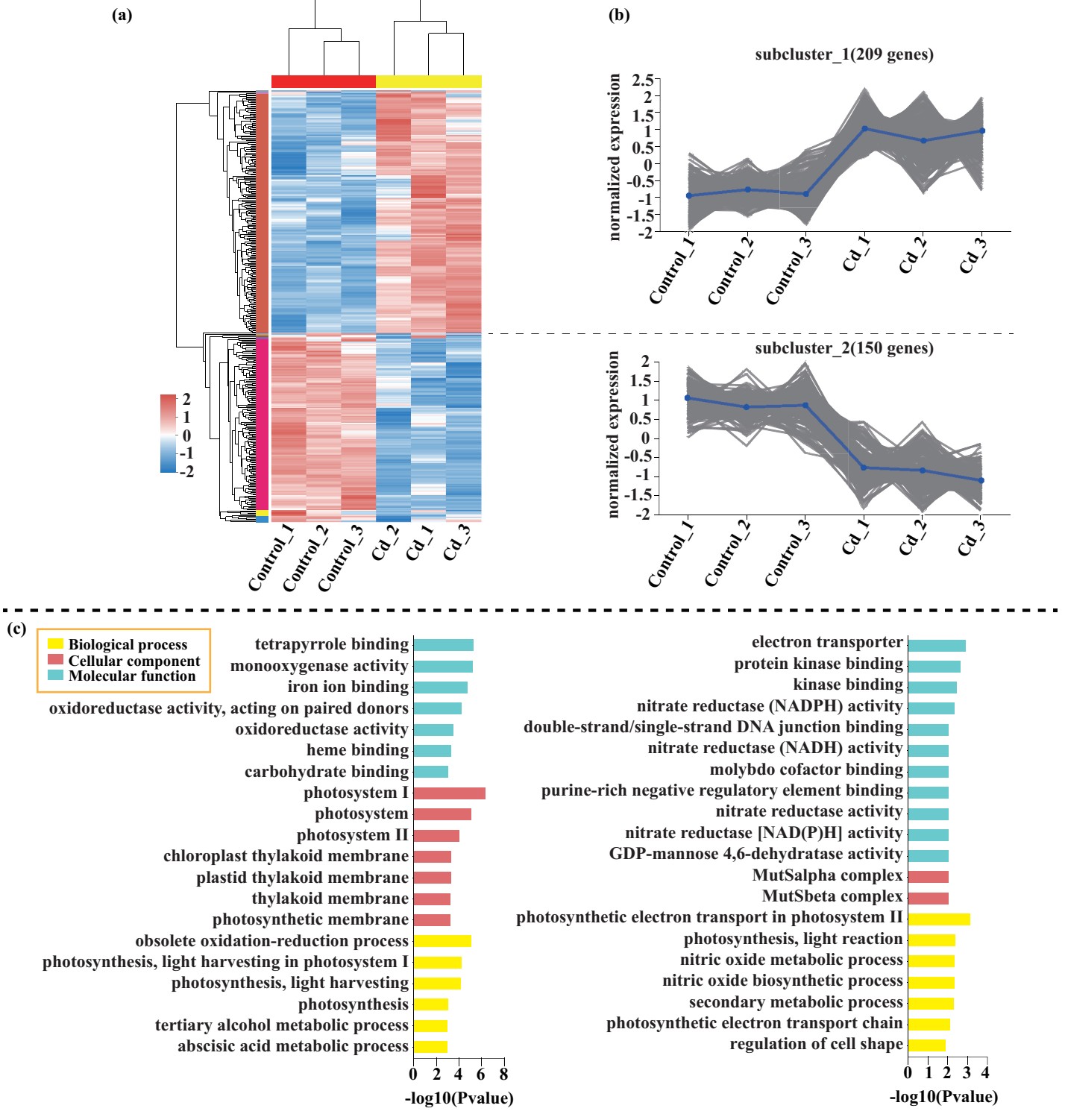

**Figure 5** Heatmap analysis (A), gene co-expression clusters (B), and GO enrichment analysis (C) of DEGs in Cd *vs.* Control group of tomato seedlings.

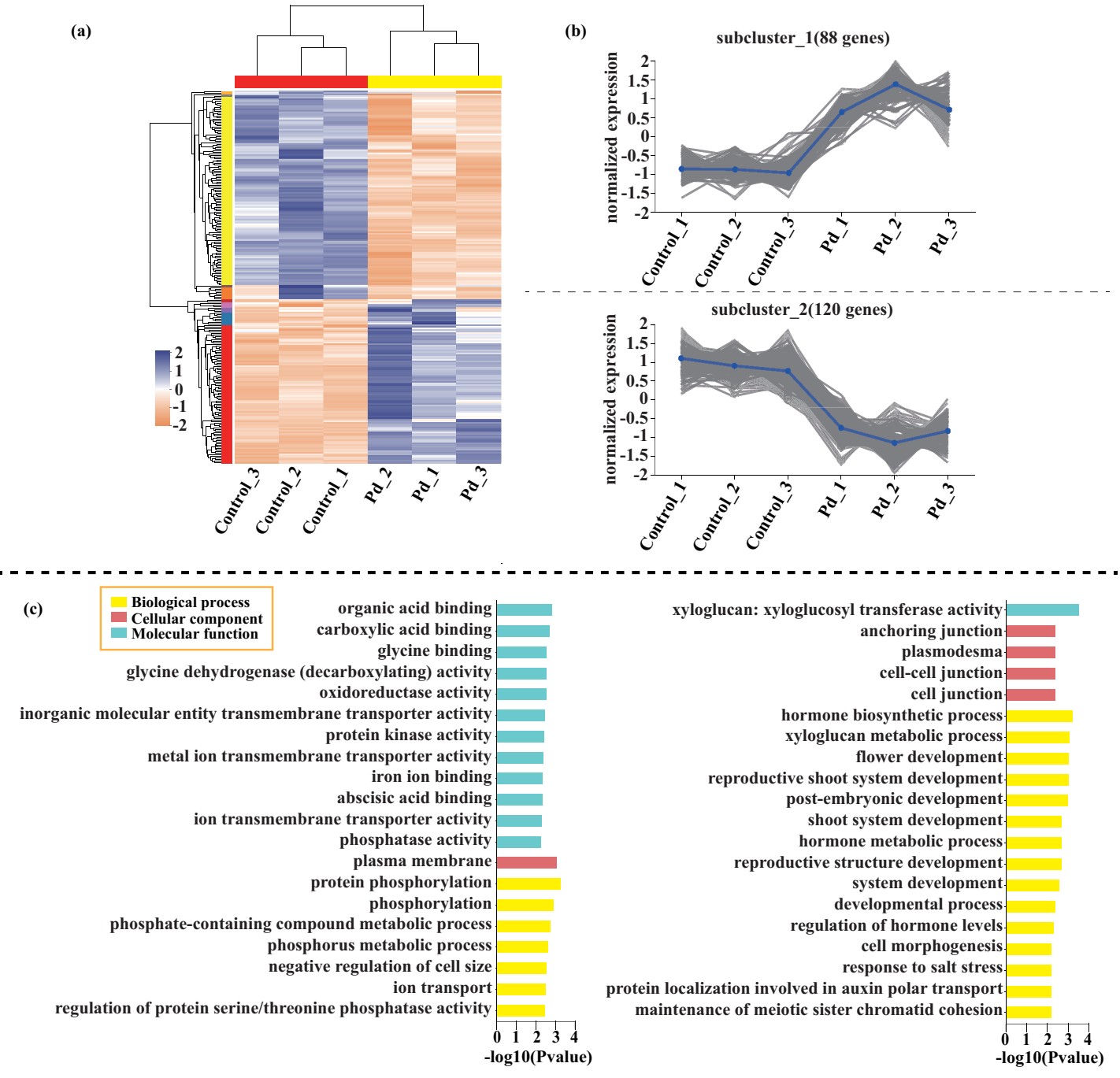

**Figure 6** Heatmap analysis (A), gene co-expression clusters (B), and GO enrichment analysis (C) of DEGs in Pd *vs.* Control group of tomato seedlings.

## Response of phenylalanine, glutathione, arginine and proline, nitrogen metabolism, and transcription factors to Cd and Pb stress

Five candidate DEGs associated with phenylalanine metabolism in the tomato seedlings under Cd, Pb, and Cd + Pb stress were identified. These genes included two *PALs* and three *HDCs*, which were upregulated in the tomato seedlings under Cd, Pb, and Cd + Pb stress

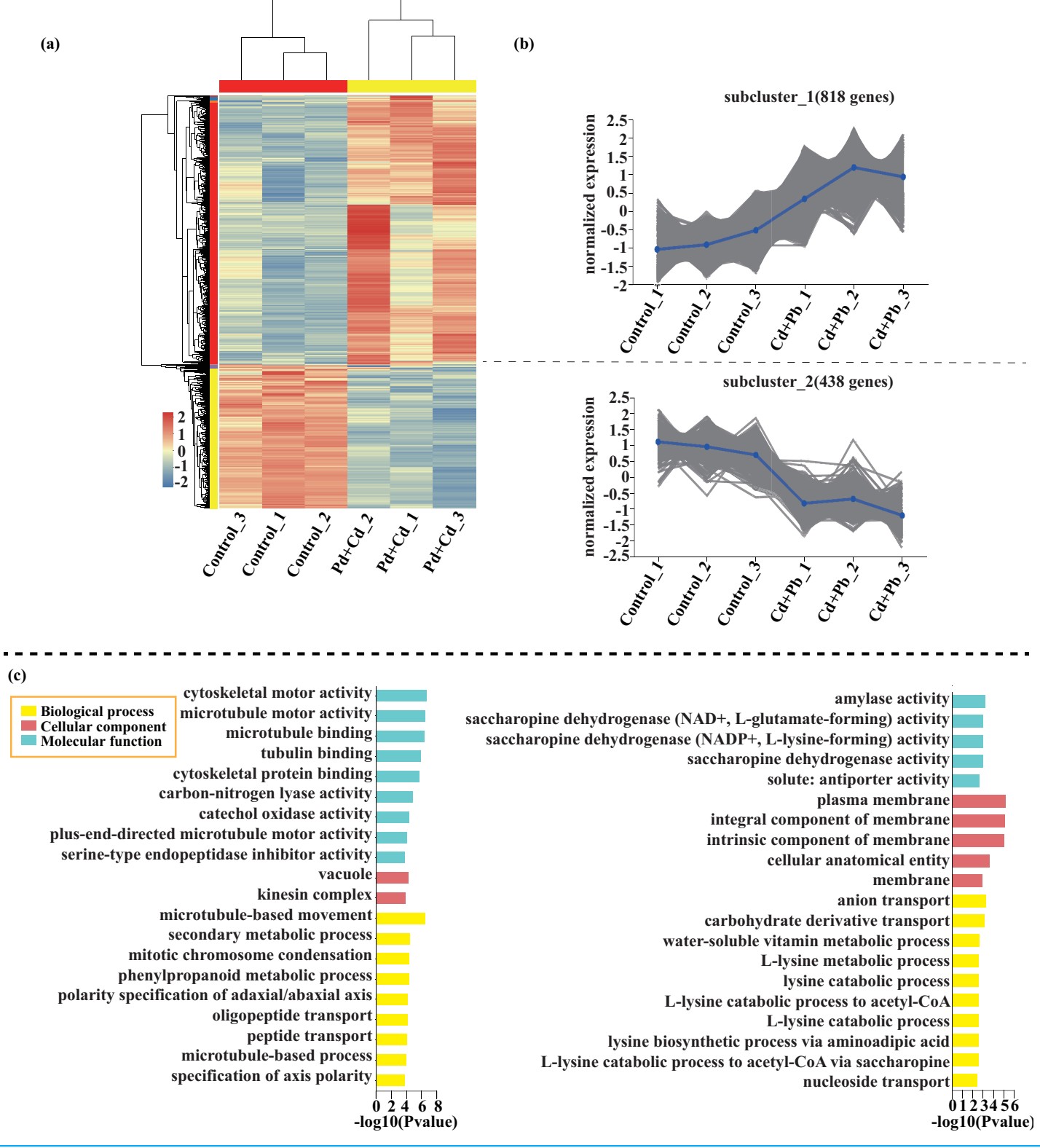

**Figure 7** Heatmap analysis (A), gene co-expression clusters (B), and GO enrichment analysis (C) of DEGs in Cd + Pd *vs.* Control group of tomato seedlings.

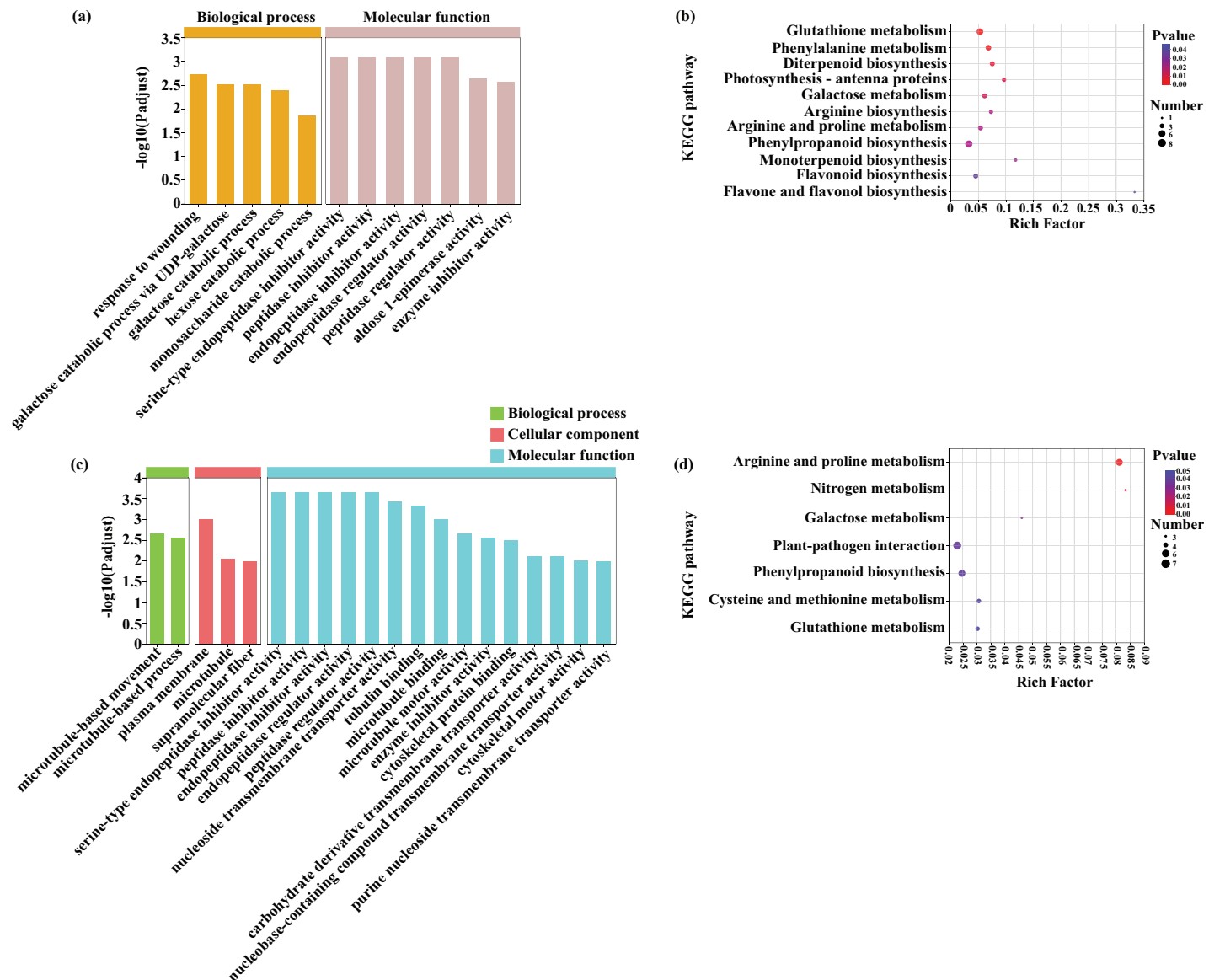

**Figure 8** GO enrichment and KEGG pathway analysis of DEGs in Cd + Pb *vs.* Pb group (A and B) and Cd + Pb *vs.* Cd group (C and D).

(Fig. 9). Eight candidate DEGs associated with glutathione metabolism in the tomato seedlings under Cd, Pb, and Cd + Pb stress were identified. These genes included three *GSTs*, *GPX*, *GGCT*, *ODC1*, *LAPs*, and *SMS*. The three *GSTs* and *GPX* were downregulated in the tomato seedlings under Cd, Pb, and Cd + Pb stress, while *GGCT*, *ODC1*, *LAPs*, and *SMS* were upregulated in the tomato seedlings under Cd + Pb stress. Six candidate DEGs associated with arginine and proline metabolism were identified in the tomato seedlings under Cd, Pb, and Cd + Pb stress. These genes included *ALDH*, *ProDH*, *P5CS*, *SMS*, *SAMDC*, and *ODC1*. *ALDH* and *ProDH* were downregulated in the tomato seedlings under Cd, Pb, and Cd + Pb stress, while *SMS*, *SAMDC*, and *ODC1* were upregulated in the tomato seedlings under Cd + Pb stress. Three candidate DEGs, *CA*, *GltS*, and *nirA*, associated with

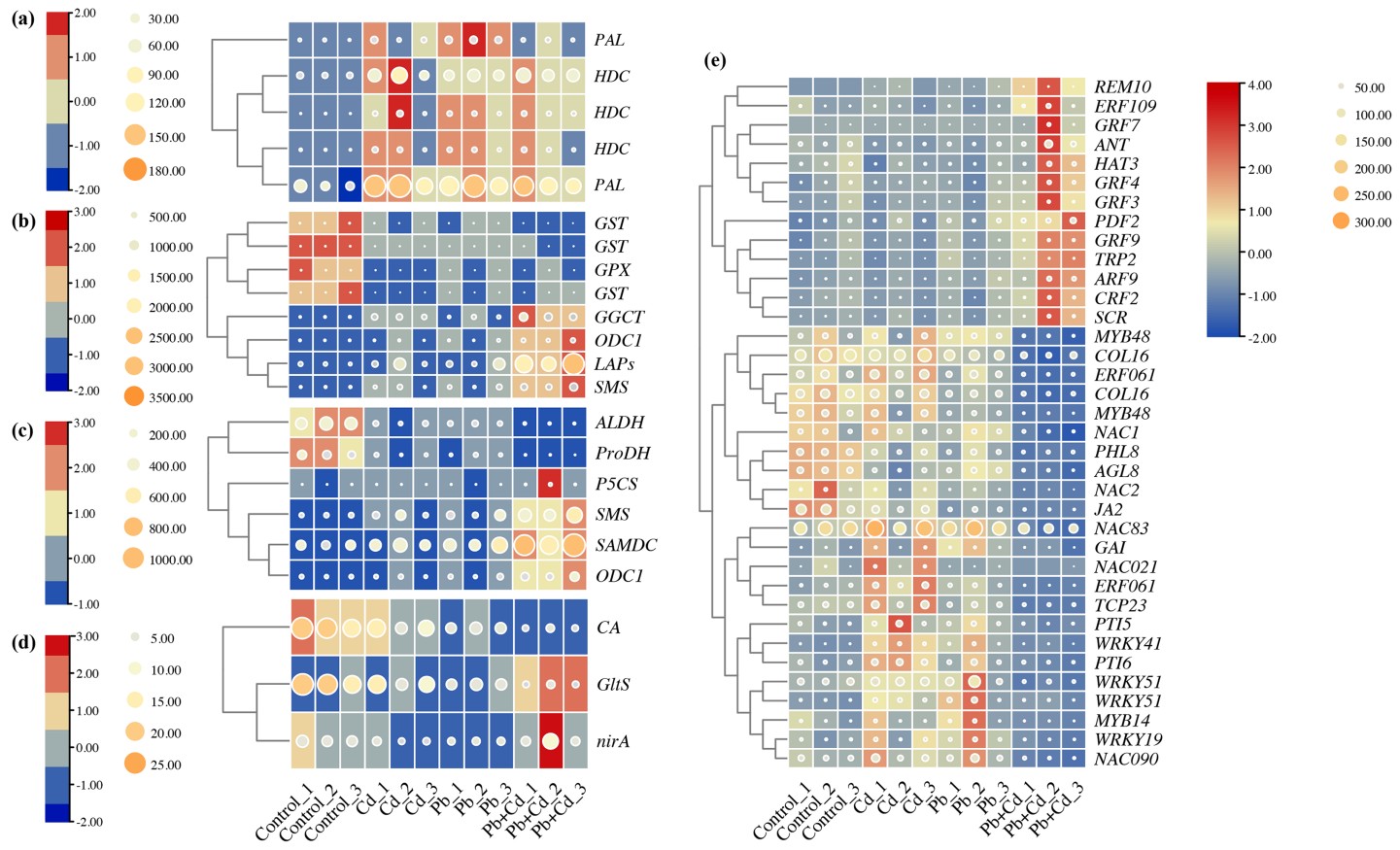

**Figure 9** Heatmap analysis of differentially expressed genes (DEGs) related to (A) phenylalanine metabolism, (B) glutathione metabolism, (C) arginine and proline metabolism, (D) nitrogen metabolism, and (E) transcription factors in the leaves of Cd- and Pb-stressed. The numbers in the heatmap represent the log2fold change values of gene expression. Higher numbers indicate greater up-regulation, while lower numbers indicate greater down-regulation of gene expression in response to Cd and Pb treatments.

nitrogen metabolism in the tomato seedlings under Cd, Pb, and Cd + Pb stress were also identified. *CA* was downregulated in the tomato seedlings under Cd, Pb, and Cd + Pb stress, while *GltS* and *nirA* were downregulated in the tomato seedlings under Cd + Pb stress. Consistently, 36 differentially expressed TFs in the tomato seedlings under Cd, Pb, and Cd + Pb stress were identified (Fig. 9). *REM10*, *ERF109*, four *GRFs* (*GRF3/4/7/9*), *ANT*, *HAT3*, *PDF2*, *TRP2*, *ARF9*, *CRF2*, and *SCR* were upregulated in the tomato seedlings under Cd + Pb stress. Two *MYB48s*, *COL16*, *ERF061*, *COL16*, *NAC1*, *PHL8*, *AGL8*, *NAC2*, *JA2* were downregulated in the tomato seedlings under Cd + Pb stress. Three *NACs* (*21/83/90*), *GAI*, *ERF061*, *TCP23*, *PTI5*, four *WRKYs* (*19/41/51-1/51-2*), *PTI6*, and *MYB14* were upregulated in the tomato seedlings under Cd and Pb stress.

## Weighted gene co-expression network

Alterations in gene transcription play a pivotal role in modulating the adaptive mechanisms of the tomato seedlings when exposed to Cd, Pb, and combined Cd + Pb. Notably, there were 453 DEGs shared between the Pb *vs.* Control, Cd *vs.* Control, and Cd + Pb *vs.* Control groups and 381 common DEGs between the Cd + Pb *vs.* Pb and Cd + Pb *vs.*
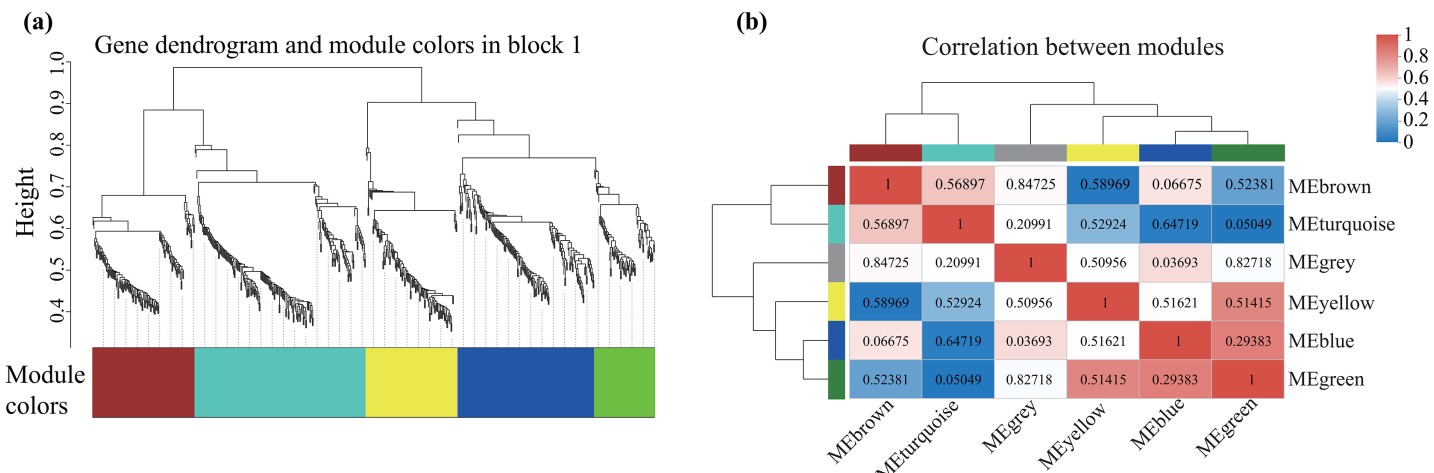

**Figure 10 WGCNA and identification of DEGs in Cd + Pb *vs.* Pb group, and Cd + Pb *vs.* Cd group.** Cluster dendrogram showing different modules (by color) of co-expressed genes as identified by the Dynamic Tree Cut algorithm and by merging modules sharing a correlation above 0.25 (A). Hierarchical clustering dendrogram (upper panel) and correlation heatmap (lower panel) of module eigengenes (ME) to examine higher-order relationships between the modules (B). Electrolytic leakage (A), proline content (B), soluble sugar content (C) and MDA (D) in leaves of different concentrations of Cd-stressed tomato seedlings. Each value is the mean ± standard error (n = 3), and the error bars represent the standard error. Bars with a different letter within a sampling date are significantly different (*P* < 0.05).           

Cd groups. A Weighted Gene Co-expression Network Analysis (WGCNA) revealed the association between diverse physiological parameters and pivotal genes within the tomato seedlings (Figs. 10–12). Gene clusters that exhibited a high correlation within the WGCNA were designated as modules and encapsulated genes with significant interrelations. Six modules, encompassing 809 genes, were differentiated (Fig. 10). Of note, 164 genes were categorized within the blue module. This study meticulously analyzed the distribution of the genes across each module from a singular gene in the grey module to 206 in the turquoise module. The correlation between the physiological metrics of tomato seedlings, and the identified modules was further elucidated through heat map visualizations, which uncovered their interdependencies. Genes within the turquoise module exhibited a positive correlation with the soluble sugar, $H_2O_2$, and levels of $O_2^{\cdot-}$ (*P* < 0.05), while those in the green module were inversely associated with the levels of soluble sugar and $O_2^{\cdot-}$ (*P* < 0.05) but exhibited a positive association with electrolytic leakage (Fig. 11). An in-depth exploration into the nexus between the levels of soluble sugar and the composition of the module was conducted by sieving transcripts in the turquoise and green modules based on the highest gene significance and module membership scores. This selection process yielded modules for the genes under scrutiny termed Modular Gene Interest (MGI). The turquoise MGI positively correlated with the levels of soluble sugar and was predominantly upregulated. Conversely, the green-MGI negatively correlated with the levels of soluble sugar and was predominantly downregulated in the tomato seedlings. This suggested a regulatory influence on the accumulation of soluble sugars. A total of 30 central genes were selected within the turquoise and blue modules based on the connectivity criterion ($k \geq$ 100.00) and edge weight $\geq$ 0.02 (Fig. 12). These findings collectively underscored the efficacy of the gene network visualization analyses in identifying the core physiological

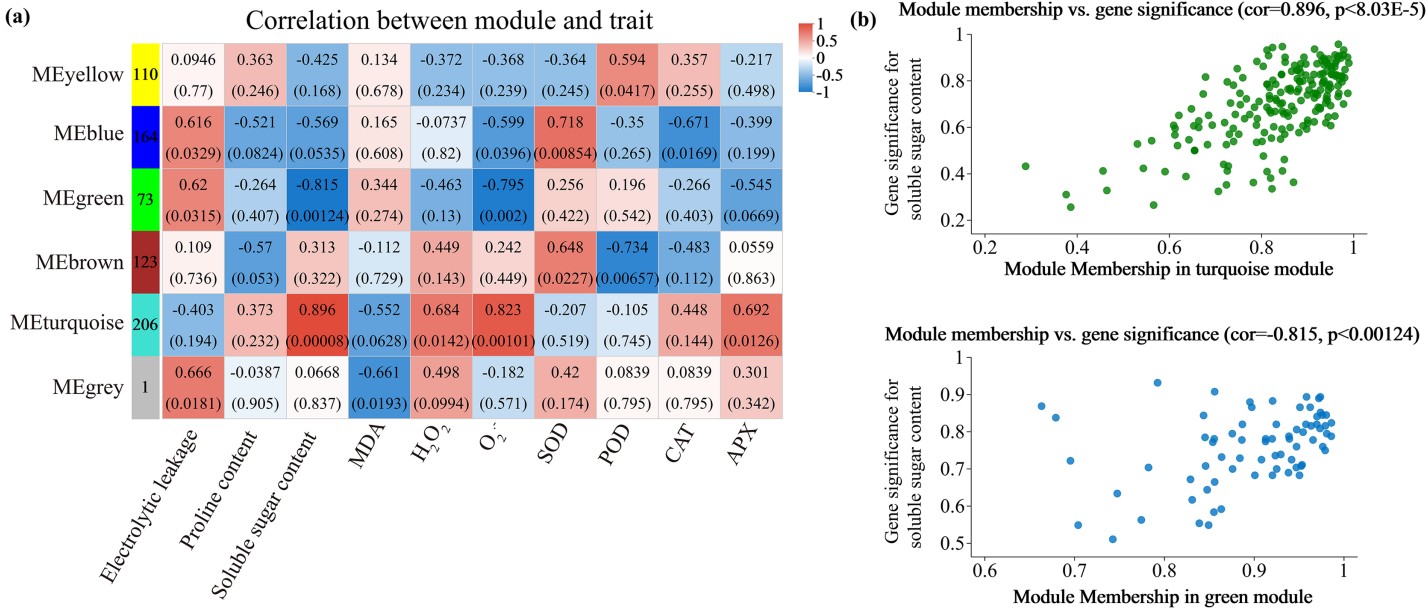

**Figure 11 WGCNA and identification of DEGs in Cd + Pb *vs*. Pb group, and Cd + Pb *vs*. Cd group.** Heatmap of physiological indicators correlations (A). Scatterplots of module membership *vs* genes significance in the brown and turquoise modules (B).

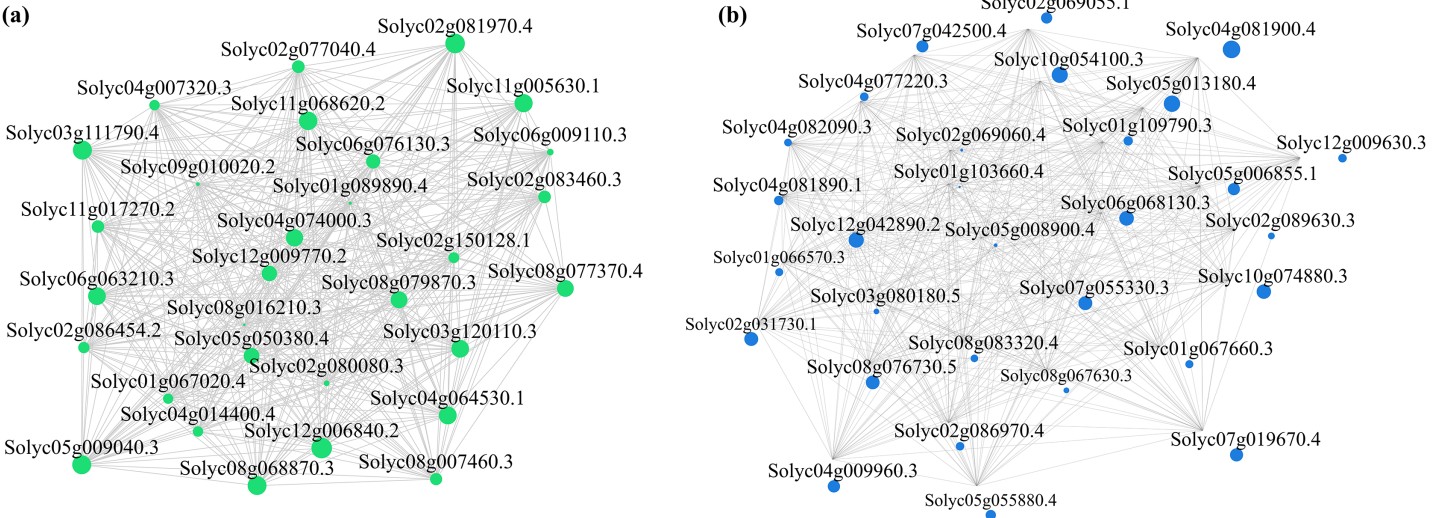

**Figure 12 WGCNA and identification of DEGs in Cd + Pb *vs*. Pb group, and Cd + Pb *vs*. Cd group.** Cytoscape representation of the module of genes of interest (MGI) within brown and turquoise modules (A). Edges with weight above a threshold of 0.02 are shown (B).

pathways and the gene clusters that modulate these pathways in tomato seedlings, thus, highlighting their potential involvement in conferring resistance to Cd and Pb stress.

## DISCUSSION

The impact of heavy metal stress on plants is reflected by their growth processes. Thus, morphological growth indicators can be used to assess the extent of the plant response to

heavy metal stress (*Jaishankar et al., 2014*). Existing studies postulate that heavy metal stress significantly affects the growth and development of plants. Adverse conditions can also induce the production of osmoregulatory substances, increase the concentration of cell sap, and reduce the water potential. This enabled the cells to absorb water from the external environment, thereby maintaining normal plant growth (*Morkunas et al., 2018*). Studies have postulated that the accumulation of proline during stress is an indicator of plant resistance (*Ghosh et al., 2022*). Stress conditions cause an increase in the content of soluble sugar because of the breakdown of starch and other sugars. The photosynthetic products are directly converted into substances of relatively low molecular weight (*Dong & Beckles, 2019*). In this study, the content of proline in the tomato seedlings in the later stages, and the content of soluble sugars in the seedlings in the early stages of Cd + Pb stress were higher than those under single stress but lower than the corresponding levels in the control seedlings. This finding suggests that Cd and Pb stress inhibits the accumulation of osmoregulatory substances, which causes the plants to be osmotically stressed. Notably, the composite stress of Cd and Pb exacerbated the toxic effects on tomato seedlings, thereby affecting their growth.

Heavy metal stress disrupts the dynamic equilibrium of the plant ROS, thereby inducing oxidative stress responses. Heavy metal stress leads to the generation of large amounts of ROS, which intensifies lipid peroxidation in the cell membrane, thus, causing membrane damage (*Mansoor et al., 2023*). Conductivity and MDA are crucial indicators that reflect the degree of membrane damage (*Dharmajaya & Sari, 2022*). Excessive amounts of ROS ($H_2O_2$, $O_2^{\cdot-}$, and peroxy radicals [$\cdot OH$]) cause oxidative damage to the structure and function of the cell membrane, which accelerates cellular senescence. Endogenous enzymatic and non-enzymatic systems of plants can eliminate ROS and prevent oxidative stress (*Nadarajah, 2020*). SOD, POD, CAT, and APX are the primary antioxidant enzymes in plants that are responsible for clearing the ROS and gradually reducing the amount of oxidative damage to the plant. Herein, Cd and Pb stress increased the conductivity and content of MDA in the tomato seedlings during the late stage of stress. Notably, the composite stress of Cd and Pb significantly reduced the levels of $H_2O_2$ compared to the single stress and reduced the levels of $O_2^{\cdot-}$ compared to the control. The activity of SOD significantly increased throughout the period of stress under different stress conditions. The composite Pb and Cd stress induced a significant increase in the activities of POD and CAT during the late stage and mid-stage of stress, respectively. Previous studies postulated that Pb stress does not alter the levels of MDA in tea (*Camellia sinensis*) (*Duan et al., 2020*). In contrast, Cd stress significantly increases the content of MDA in watermelon (*Citrullus lanatus*) (*Khan et al., 2021*), chili pepper (*Capsicum annuum*) (*Kaya, 2020*), and maize (*Zea mays*) (*Zhang et al., 2017a*) seedlings. These findings suggest that the exposure of the tomato seedlings to Cd and Pd stress can enhance the activities of SOD, POD, and CAT to scavenge excess $H_2O_2$ and $O_2^{\cdot-}$, thereby reducing the degree of membrane lipid peroxidation, which protects the membrane structure. Moreover, the Cd and Pb composite stress induces mechanisms of stress response in tomatoes, which highlights a certain level of tolerance to the composite stress of Cd and Pb.

The effects of Cd and Pb stress on plant metabolic processes are multifaceted and primarily involve the interference of the physiological and biochemical processes, cell structure, and metabolic pathways (*Collin et al., 2022*; *Wei et al., 2023*). Studies have postulated that the Cd and Pb stress primarily affect the organic acids, amino acids, lipids and energy metabolism of plants (*Haider et al., 2021*; *Asare, Száková & Tlustoš, 2022*; *Cuypers et al., 2023*). In this study, the transcriptome analysis revealed that the Cd and Pb stress primarily affected the metabolic process. Plants exhibit various physiological and molecular regulatory mechanisms to adapt to environmental abiotic stresses (*Nawaz et al., 2023*). Phenylalanine metabolism is an important plant regulatory mechanism that plays a crucial role in the responses of plants to stress (*Anzano et al., 2022*). Plants often increase their rate of biosynthesis of phenylalanine under abiotic stress (*Sharma et al., 2019*). This increase is potentially achieved by enhancing the activities of enzymes involved in the biosynthesis of phenylalanine, such as phenylalanine ammonia-lyase (PAL) (*Zhang et al., 2017b*). This increased biosynthesis potentially contributes to providing sufficient substrates for the biosynthesis of secondary metabolites by plants to manage environmental stress. In this study, the levels of expression of the *PALs* and *HDCs* involved in phenylalanine metabolism were significantly upregulated under combined Cd and Pb stress. Previous studies have reported that four *PALs* are involved in the plant growth, development, and response to environmental stress in *Arabidopsis thaliana* (*Huang et al., 2010*). Some studies suggest that the levels of expression of *HDCs* increase in plants under drought stress and are thus, possibly associated with the adaptation of plants to drought stress (*Oguz et al., 2022*). Glutathione plays a crucial role in protecting the cells from the effects of biotic and abiotic stresses (*Zhou et al., 2022b*). It is the primary substance in the cells that resists free radicals and ROS, thus, protecting the cells from damage by various exogenous substances and carcinogens. A series of changes occur in glutathione metabolism, such as guiding DNA repair, protein synthesis, and amino acid transfer in the plants under abiotic stress, to enhance their adaptation to external environmental pressures (*Hameed et al., 2014*). In this study, the levels of expression of the *GGCT*, *ODC1*, *LAPs*, and *SMS* genes involved in glutathione metabolism were significantly upregulated under the combination of Cd and Pb stress. Plants require more γ-glutamyl transferase to participate in the glutathione biosynthetic pathway to enhance resistance to oxidative stress (*Hasanuzzaman et al., 2017*). Studies have postulated that the level of expression of *ODC1* increases in plants under abiotic stress (*Cai et al., 2017*). ODC participates in the polyamine biosynthetic pathway and maintains the stability of the levels of polyamines within the cells when managing stress (*Kahana, 2009*). SMS is a vital enzyme in plants that is involved in the biosynthesis of polyamines, such as spermine (*Roy & Wu, 2002*). The level of expression of *SMS* increases in plants under abiotic stress (*Thu-Hang et al., 2002*). SMS is a vital enzyme in the biosynthesis of spermine, which plays an important role in the resistance of plants to stress. Thus, an increase in the level of expression of spermine helps to enhance the adaptation of plants to stress (*Kasukabe et al., 2004*; *Wen et al., 2008*). Abiotic stress potentially increases the levels of expression of *LAPs* in plants. *LAPs* participate in the regulation of amino acid metabolism when managing stress and maintaining the nitrogen metabolic balance (*Krasensky & Jonak, 2012*). The combination

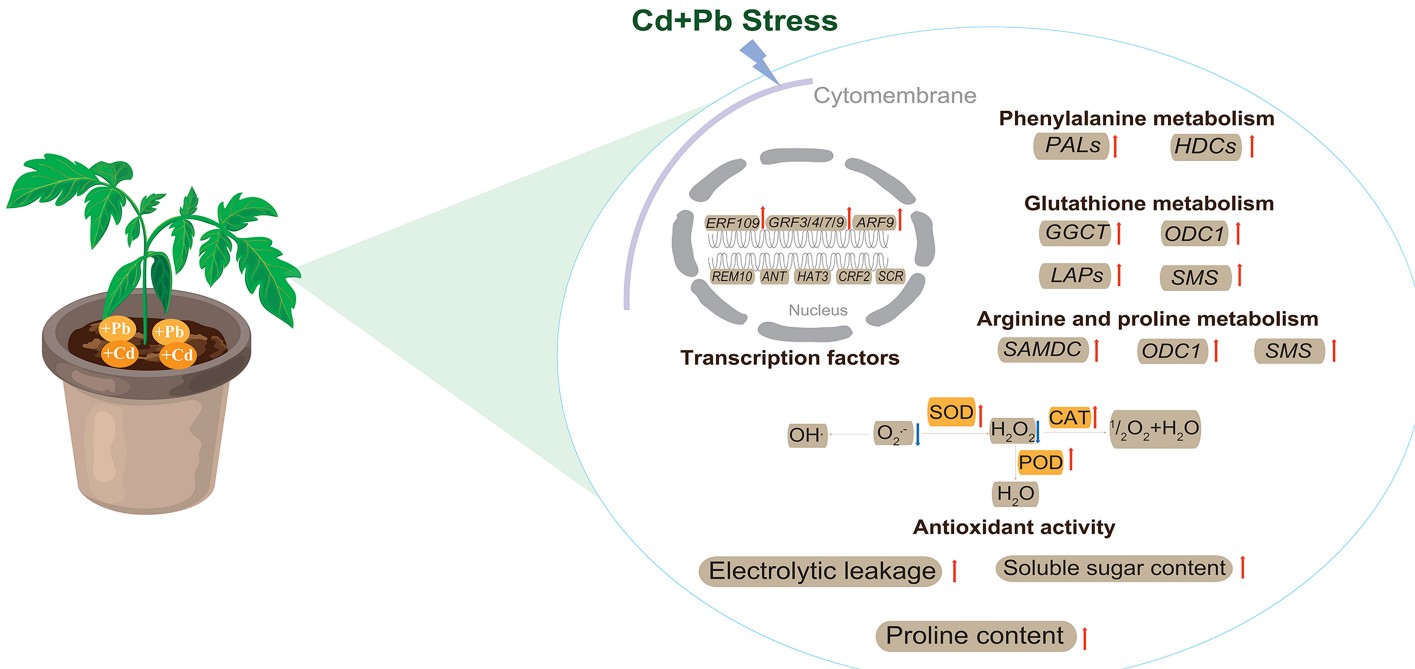

**Figure 13 The model pattern of tomato seedling response to Cd and Pb stress.** Red arrows indicate up-regulation, while blue arrows indicate down-regulation.

of Cd and Pb stress led to a significant upregulation of *SAMDC*, *ODC1*, and *SMS* in the arginine and proline metabolic pathways in the leaves of tomato seedlings. SAMDC is one of the vital enzymes involved in the biosynthesis of polyamines in plants (*Roy & Wu, 2002*). S-adenosylmethionine (SAM) is a precursor to this biosynthesis of polyamines. Increasing the expression of *SAMDC* increases the biosynthesis of polyamines, thereby enhancing the resistance of plants to stress (*Sauter et al., 2013*). In rice (*Oryza sativa*), the overexpression of *SAMDC* increases the levels of polyamines, thereby enhancing its tolerance to NaCl stress (*Roy & Wu, 2002*). Similarly, the overexpression of *SAMDC* in tomato increases its tolerance to alkali stress (*Gong et al., 2014*). The overexpression of *SAMDC* enhanced the tolerance to alkali in transgenic tomato (*Cheng et al., 2009*). Herein, the expression of the *SAMDC* gene was potentially one of the response mechanisms of tomato seedlings to Cd and Pb stress.

TFs regulate the physiological and biochemical processes of plants, participate in signaling pathways, and enhance plant stress resistance by modulating the expression of stress-responsive genes (*Meraj et al., 2020*; *Yoon et al., 2020*). In this study, Cd and Pb composite stress significantly upregulated *ERF109*, *ARF9*, *GRF3*, *GRF4*, *GRF7*, and *GRF9*. *ERF109* is a crucial response factor regulated by conditions of metal stress. It potentially aids plants to adapt to such environments by modulating the expression of the genes related to metal stress. *ERF109* can serve as a "master switch" mediator under high salt stress that promotes plant growth and adaptation under adverse conditions by regulating the expression of genes involved in the biosynthesis of phenylalanine, tyrosine, and tryptophan, tryptophan metabolism, and plant hormone signal transduction pathways

(*Bahieldin et al., 2018*). *ARF9* negatively regulates cell division during early tomato fruit development and is primarily involved in the auxin signaling pathway. It responds to metal stress by interacting with other stress response pathways or regulating auxin-related genes (*De Jong et al., 2015*). Growth-regulating factors (*GRFs*) are highly conserved TFs in plants that regulate various developmental processes and plant responses to biotic and abiotic stress. Previous studies have postulated that *OsGRF4* integrates N assimilation, C fixation, and growth (*Li et al., 2018*). *GRF1* and *GRF3* play vital roles in plant growth and development, the biosynthesis of phytohormones and their signaling, and the cell cycle (*Piya et al., 2020*). *SsGRF7* positively regulates the size and length of rice leaves by modulating cellular size and plant hormones (*Wang et al., 2024*). A. *thaliana* wild-type, *GRF9*-deficient mutant (*grf9*), and *GRF9*-overexpressing (OE) plants were treated with polyethylene glycol (PEG) to induce mild water stress (*He et al., 2015*). In this study, Cd and Pb enhanced the levels of expression of *GRF3*, *GRF4*, *GRF7*, and *GRF9*, which highlights their potential involvement in plant growth and resistance to metal stress by modulating the levels of gene expression. Although the specific mechanisms of these TFs in the responses to metal have not yet been fully elucidated, the findings of this study provide important clues to further unravel the molecular regulatory network of plant metal stress responses. They also offer valuable references and guidance for the future enhancement of plant resistance to metal stress using genetic engineering approaches.

## CONCLUSION

This study utilized biochemical and transcriptomic analyses to explore the stress response and mechanisms of the tolerance of tomatoes under combined Cd and Pb treatment (Fig. 13). Tomato seedlings responded to combined Cd and Pb stress by significantly increasing electrical leakage, the contents of proline and soluble sugar, and enhancing the activities of key antioxidant enzymes, such as SOD, POD, and CAT. These physiological responses may collectively improved the antioxidative capacity and contribute to the tolerance of the tomato seedlings to stress. A transcriptomic analysis revealed that the DEGs are involved in crucial metabolic pathways, including phenylalanine, glutathione, arginine, and proline metabolism. These pathways could enhance the ability of the cell wall to block and accumulate Cd and Pb ions, thereby effectively mitigating the toxic effects of these heavy metals on the plants. The findings of this study provide new insights into the growth, development, and molecular response mechanisms of tomatoes under combined metal stress. These findings highlight the complex interplay between the physiological and molecular processes in the tolerance to plant stress, which offers potential strategies to improve the resilience of plants in contaminated environments.

### Funding

This work was funded by the National Natural Science Foundation of China (32202476), the Natural Science Foundation of Guangdong Province (2024A1515012858), the Science and Technology Plan Project of Zhanjiang (2022A01030), the Lei Yang Academic Posts

Programmer of Lingnan Normal University (2022), the Scientific research team project of Lingnan Normal University (LT2201), the School-level Talents Project of Lingnan Normal University (ZL2033), and the scientific research promotion of key construction discipline in Guangdong (2022ZDJS080). The funders had no role in study design, data collection and analysis, decision to publish, or preparation of the manuscript.

## Grant Disclosures

The following grant information was disclosed by the authors:
National Natural Science Foundation of China: 32202476.
Natural Science Foundation of Guangdong Province: 2024A1515012858.
Science and Technology Plan Project of Zhanjiang: 2022A01030.
Lei Yang Academic Posts Programmer of Lingnan Normal University: 2022.
Scientific Research Team Project of Lingnan Normal University: LT2201.
School-level Talents Project of Lingnan Normal University: ZL2033.
The Scientific Research Promotion of Key Construction Discipline in Guangdong: 2022ZDJS080.

## Competing Interests

The authors declare that they have no competing interests.

## Author Contributions

- Yan Zhou conceived and designed the experiments, performed the experiments, prepared figures and/or tables, authored or reviewed drafts of the article, and approved the final draft.
- Jinyu Fu conceived and designed the experiments, authored or reviewed drafts of the article, and approved the final draft.
- Yuqi Ye performed the experiments, authored or reviewed drafts of the article, and approved the final draft.
- Qibo Xu analyzed the data, prepared figures and/or tables, and approved the final draft.
- Jinjie Liang analyzed the data, authored or reviewed drafts of the article, and approved the final draft.
- Yanyan Chen analyzed the data, prepared figures and/or tables, and approved the final draft.
- Yuxing Mo conceived and designed the experiments, prepared figures and/or tables, and approved the final draft.
- Kaidong Liu performed the experiments, authored or reviewed drafts of the article, and approved the final draft.

## Data Availability

The raw sequence data are available at NCBI: PRJNA1085857.

## Supplemental Information

Supplemental information for this article can be found online at http://dx.doi.org/10.7717/peerj.18533#supplemental-information.

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
