# Peer review of "Physiological and molecular response mechanisms of tomato seedlings to cadmium (Cd) and lead (Pb) stress"

_PeerJ, doi:10.7717/peerj.18533_

## Round 0.1 · original submission · Major Revisions

Several reviewers have evaluated your paper, and considering their comments and my personal judgement, I feel that your paper needs to be revised and improved.

·

Basic reporting

No comments

Experimental design

No comments

Validity of the findings

The manuscript is very well-written and captivates the audience. It makes the readers eager to reach the next section. Congratulations to the authors for such a finely written scientific piece!
I do have a comment: in lines 133 and 134, the authors mention the amount of metal used for the experiments. It would be beneficial to include in the manuscript the reason behind that choice of concentration.

·

Basic reporting

Manuscript entitled “Physiological and molecular response mechanisms of tomato seedlings to Cd and Pb stress” assessed explore the physiological and molecular response mechanisms of tomato seedlings to Cd and Pb stress. The results of this study showed that tomato seedlings can respond to osmotic and oxidative stress caused by Cd and Pd stress by enhancing their osmoregulation capacity and resistance to the reactive oxygen species scavenging system. The findings are significant, the experimental methods and results are acceptable. However, to improve the quality of the manuscript, there are some essential problems should be addressed by authors, which are listed below.
1. Some writing problems in the article are listed below:
L39. Keywords should be in alphabetical order.
L144. A reference should be added.
L179. The initial letter of “analysis” should be lower case.
2. Figure or Fig.? There should be unity in the manuscript.

Experimental design

GOOD

Validity of the findings

GOOD

Reviewer 3 ·

Basic reporting

The manuscript describes the physiological and molecular responses of tomato seedlings to cadmium (Cd) and lead (Pb) stress. It highlights the increasing environmental concern of Cd and Pb contamination and their detrimental effects on plant growth. The study treated tomato seedlings with 50 mg/L Cd, 100 mg/L Pb, and a combination of both, examining impacts on growth, antioxidant systems, and secondary metabolic pathways. Findings showed increased soluble sugar and proline contents, elevated electrical leakage, maintained malondialdehyde (MDA) levels, enhanced antioxidant enzyme activities, and reduced H2O2 and O2·- contents. Transcriptomic analysis revealed significant changes in gene expression related to phenylalanine, glutathione, arginine, proline, and nitrogen metabolism, with transcription factors like ERF109 and several GRFs being notably regulated. The study demonstrates that tomato seedlings mitigate osmotic and oxidative stress through enhanced osmoregulation and reactive oxygen species (ROS) scavenging, thereby maintaining normal growth. However, the authors need to improve their language, and also have some other problems that need revision.

Experimental design

NA

Validity of the findings

1. There are no error bars shown in Table 1. Please correct the figure legend. It would be clear if the authors could subdivide the table into three sections each having one time point.
2. For figure 1, it would be better to have all data points displayed on the figure. The authors should make clear that statistics were separately conducted at each time point. The same for Figure 2 and 3.
In the Figure 3c, the data for 10d seems not correct. Please add all data points to the figure.
3. Figure 4 legend needs to be finished.
4. Figure 7 legend needs improvement, for example, what do all the numbers mean in the figure. The phrase “response to” is confusing. The authors did gene expression analysis in response to the treatments and observed DEGs. Please make the figure legend clear.

Additional comments

NA

Reviewer 4 ·

Basic reporting

no comment

Experimental design

no comment

Validity of the findings

no comment

Additional comments

I have reviewed this manuscript which has been submitted to the journal PeerJ. The main question addressed by the research is how tomato seedlings respond physiologically and molecularly to combined cadmium (Cd) and lead (Pb) stress. The study aims to examine the phenotypic, physiological, and transcriptional changes in tomato plants upon exposure to these heavy metals and identify the genes responsive to Cd and Pb stress. After checking the manuscript and comparing it with the current literature, I realized that the manuscript needs substantial improvements as the following comments:

This research focus could be better justified in terms of originality, considering that there are already many studies on the same research topic. The authors do not even cite many articles on this same research topic and field. Additionally, the justification for the experimental and methodological settings, including the choice of tomato genotype, heavy metal stress treatments, and exposure times, is not clear to me. Indeed, incorporating more tomato genotypes could have added robustness to the hypothesis, providing insights into this plant species in a more representative and comprehensive manner. Furthermore, more phenotypic, biomass, and dry mass data are needed to better elucidate the hypothesis focused on in this present study. The authors could also provide insights based on proteomics, considering that proteins are more related to the plant phenotype than transcripts. Especially considering the research topic of so many existing articles on this theme in tomato plants, proteins are essential to bring novelty. Other considerations include: The English language should be improved for clarity; The supplemental files need more descriptive metadata identifiers to be useful to future readers; The data analysis could be improved by providing more detailed and in-depth statistical analysis; recent articles (from 2023 and 2024) could be better incorporated. Moreover, the manuscript would benefit from a more thorough discussion on the potential applications of the findings in agricultural practices and environmental management. Additionally, it would be beneficial if the discussion on antioxidant balance were aligned with a better scientific comparison of the literature on this research topic and included metabolomic analysis to better elucidate non-enzymatic antioxidant metabolism.

·

Basic reporting

Dear authors
The researchers investigated the physiological and molecular response of tomato under Cd and Pb stress conditions. There is little new information on the subject of this investigation. The following adjustments are needed:

Abstract
 The researchers must state the issue in a single sentence and explain why they chose this strategy for conducting the investigation.
 The experimental design and its component should be defined
 All abbreviations should be written in complete name
 Some scored data also should be included
 The conclusion is too long and should be converted into a single line
 The authors should include a single line about future prospects at the end of the abstract.
Keywords
 The terms in the title should not be used as keywords. As a result, the keyword structure should be modified.
Introduction
 This section is too long and should be shortened
 It is better to use the recent references
 Detail information about the sources of contamination of soil by Cd and Pb should be added
 Detail information about the tomato plant should be added. Why did the researchers select this plant species for study this phenomenon?
 The authors should provide a few paragraphs outlining the knowledge gap that their research addressed in addition to the hypothesis statement.
 In addition, the authors should provide a declaration of uniqueness in their conclusion.
 In comparison to earlier studies, what new research-related linkages or activities have the authors found in this one?
 The general and specific objectives should be included.

Materials and Methods
 The quantity of tissue should be defined for all physicochemical properties
 All abbreviation should be written in detail name
 All procedure should be supported by the references
 The procedure of Electrical leakage should be detailed
 The DNA extraction method should be detailed
 What are the criteria of selecting of these concentrations of Cd and Pb?
 The units of growth parameters should be provided

Results and discussion
 All captions should be improved. The labelling by different letters should be defined in the captions.
 All abbreviations should be defined in the captions of tables and figures
 The discussion lacked conviction. The commentary is inadequate and needs to be revised because most of the sentences repeat the findings rather than providing an analysis of the information. The relevance of each study's findings to the authors' own should be discussed. The authors should interpret the mechanism effects of Cd and Pb on the growth, physiological, molecular and biochemical responses.
Conclusion
 The authors ought to provide a summary of the most significant discoveries because this portion is presented in an easily readable manner.
 More investigation should be done in this field in future studies.

Experimental design

The type of ANOVA should be defined

Validity of the findings

It is better to show the data as the increasing or decreasing percentage compared to control plants

---

## Round 0.2 · Minor Revisions

Your article is basically ready to be accepted. However, after discussion by the editorial board and the Section Editors, there are some minor issues that still need to be revised:


1. “The Conclusions are not fully justified. Without genetic or other manipulations of the implicated pathways it is not proven that the implicated Without genetic or other manipulations of the implicated pathways it is not proven that the implicated pathways are the mechanisms(s) for stress tolerance.
So the places in the text where the conclusions are made can be adjusted. Such as: Change line 38 "...enhanced the osmoregulatory and antioxidant defense systems in tomato seedlings, which contributed to their tolerance to heavy metal stress" to "...enhanced the osmoregulatory and antioxidant defense systems in tomato seedlings, which MAY CONTRIBUTE to their tolerance to heavy metal stress", line 469 from "These physiological responses collectively improved the antioxidative capacity and contributed " to "These physiological responses MAY COLLECTIVELY IMPROVE the antioxidative capacity and CONTRIBUTE", and lines 471-473 from "These pathways were found to significantly enhance the ability of the cell wall to block and accumulate Cd and Pb ions, thereby effectively mitigating the toxic effects of these heavy metals on the plants" to "These pathways COULD enhance the ability of the cell wall to block and accumulate Cd and Pb ions, thereby effectively mitigating the toxic effects of these heavy metals on the plants"
2. The labeling font in the figures is too small in 100% view, the font size in the figures can be adjusted larger, thus improving readability.

The paper can be accepted after completing the above modifications.

·

Basic reporting

No comment

Experimental design

My inquiries have been appropriately addressed, and there are no further additions to make.

Validity of the findings

No comments

Reviewer 3 ·

Basic reporting

Given the thorough incorporation of point-by-point suggestions throughout the manuscript, I recommend its acceptance for publication. Thanks!

Experimental design

NA

Validity of the findings

NA

Additional comments

NA

---

## Round 0.3 · accepted · Accept

Congratulations, your paper was accepted.